# Hoop House and Field Evaluation of Tigernut (*Cyperus esculentus* L. var. *sativus* Boeck) Selections in New Jersey, USA

**DOI:** 10.3390/plants11070897

**Published:** 2022-03-28

**Authors:** Albert O. Ayeni

**Affiliations:** Department of Plant Biology, Rutgers University, 59 Dudley Road, New Brunswick, NJ 08901, USA; aayeni@scarletmail.rutgers.edu

**Keywords:** tigernuts, hoop house production, field production, New Jersey, USA

## Abstract

Tigernut or ‘chufa’ (*Cyperus esculentus* L. var. *sativus*) is gaining popularity in the United States as a high energy tuber crop known for sweet and chewy taste, 40–45% gluten-free digestible carbohydrate, high dietary fiber content, healthful fatty acid profile (73% monounsaturated, 18% saturated, 9% polyunsaturated—similar to olive oil), high oleic acid, and high P, K, and vitamins C. E. Tigernut tubers were obtained from specialty crop markets in central NJ and purchased online from commercial distributors as propagules for transplants for hoop house and field production studies. Nine tigernut selections were also evaluated under NJ hoop house culture conditions for growth habit and in the field for adaptation and productivity We concluded that tigernut production is feasible in NJ based on the results of these experiments. The growth patterns of three selections (GH, MV and SK) were studied and characterized. Foliage growth was similar in the three selections. Plant height ranged from an average of 90 cm in GH to 110 cm in MV and SK; side shoot production capacity ranged from 13 shoots per propagule in GH to 20 or more in MV and SK over 14 weeks. Over 99% of tubers in MV and SK were located within the upper 5 cm of the growth media (Pro-Mix BX brand) but tubers of GH were observed at greater soil depths (~20 cm). Tubers varied from spherical (round) in shape in GH and SK to oblong (elongated) in MV. In the field the best growth and tuber yields from NG3 and T-USA selections were obtained under black or white-over-black plastic mulch in conventionally managed plots. Tubers showed high levels of Fe (168–218 ppm) and Zn (39–50 ppm) implying that they should be a good source of these essential elements in human diet. Studies also showed that the tigernut tuber cannot survive the cold winter months in the field in NJ, therefore minimizing the fear of “tigernut invasion” of agronomic fields in NJ and similar agroecosystems.

## 1. Introduction

Tigernut or ‘chufa’ (*Cyperus esculentus* var. *sativa*) is a tropical sedge that produces edible tubers commercially traded as nuts. The tubers are chewy and have a sweet taste with a nutty flavor like coconut or almond [1,2]. Common names vary widely among cultures around the world. Both the plant and the tubers are called several names including aki-Hausa, atadwe, aya, chufa, earth or ground almond, earth chestnut, earth nut, edible galingale, edible rush, ofio, rushnut, tigernuts, tiger nutsedge, yellow nutgrass, yellow nutsedge, zulu nut, etc. [1,2,3]. Tigernut is a perennial that reproduces asexually through tuberous growth but can also reproduce sexually, albeit more rarely. Unlike the weedy form of the species known as yellow nutsedge in North America, which flowers readily in the field, the cultivated tigernut rarely flowers under New Jersey (NJ) conditions [personal observations]. Plant height ranges from 30–110 cm depending on soil fertility conditions and the plant thrives in moist loamy, sandy loam, and loamy sand soils (unpublished reports).

Tigernut has been cultivated since ancient times. It has been grown in North Africa, especially Egypt and the surrounding areas, for centuries [4]. The tubers have been found in the tombs of Pharaohs of pre-dynastic times, about 6000 years ago [5], and are thought to be a part of the diet of our Paleo-ancestors [6]. A recent Oxford University publication reported that tigernuts were indeed a part of the diet of early man around the Pleistocene Epoch and consisted of about 80% of the diet [7].

The tubers have moderate protein content (6–9%), high in lipids (28–35%) [8,9], and high in energy (400 kcal/100 g) [8] compared to sweet potatoes (100 kcal/100 g), another tropical tuber crop species. Tigernut-tubers can be ground into flour to be used in baking products and make an excellent wheat flour substitute for those with Celiac disease, or those seeking a gluten-free diet [1,10,11]. The tigernut oil shares a similar fatty acid profile to olive oil with high (up to 73%) monounsaturated fatty acids [12,13]. Oleic acid is the most abundant fatty acid in tigernut tubers and has been implicated in the reduction of heart disease, diabetes, and cancer [14,15,16]. Vanillin has been found in roasted tigernut oil and is a favorable and marketable quality as it adds to the overall aroma of the oil [13] Tigernut oil could serve as a natural alternative source of vanillin or aromatic food flavoring [17]. Tubers are also high in vitamins C and E [18]. In addition to all these benefits, tigernuts also contain more than the adult FAO/WHO requirements for daily protein intake [19] and more than the requirements for 17 out of 20 amino acids. Tigernuts have low amounts of (or completely lack) the essential amino acids asparagine, glutamine, and tryptophan [19]. In West and North Africa and in Spain, tigernut tubers are fermented and squeezed to produce the “*horchata de leche*” or “*horchata de chufa*”, a popular beverage in Spain and North Africa. Recent review papers and research articles described the possible use of tigernut tuber as a source of a natural anticancer drug [20] and as an aphrodisiac [21].

The global market for tigernut has increased significantly since 2015. Among the top 10 importing countries, importation of tigernut on average increased by 27.9% between 2015 and 2020, with an overall import value of $1.82 B in 2020 [22]. The United States, China, and Germany are the three leading importers of tigernuts with each country having an import share of 30.7, 16.5, and 8.7%, respectively [22]. The United States spent $660.5 M in 2020 on tigernut importation (Table 1). To combat the ongoing shortages from soybean importation for vegetable oil, in 2019 Cheng [23] reported that China may shift to tigernut production as a substitute for soybean.

In the United States, wild tigernut, popularly known as yellow nutsedge, is considered a difficult weed to manage in vegetable and field crops as well as in home or recreational lawns and turf [24,25,26]. The weedy characteristics of yellow nutsedge make it highly productive and desirable to be considered as a biofuel alternative or for other applications. There is limited cultivation in some southern states primarily directed at feeding the tubers to turkeys [27] or using them as bait for fishing [28,29]. The several good qualities of tigernuts and the rising economic value on the global market, make it a highly desirable crop to integrate into the agricultural economy of NJ and the mid-Atlantic for maximum benefits. The objective of this paper is to highlight the results of studies conducted at Rutgers University, New Brunswick, NJ, USA between 2008 and 2021 to evaluate the performance of tigernut selections in the hoop house and their adaptation to field cultivation in NJ.

## 2. Materials and Methods

### 2.1. Tigernut Tuber Sources

Between 2008 and 2019, tigernut tubers obtained from African markets around New Brunswick, NJ and commercial stores in New York (NY) and NJ were raised in hoop house nurseries to compare growth and productivity. Tubers were separated based on size, color, and shape and identified based on tuber source. Table 2 describes the tigernut selections used in the hoop house and field studies between 2008 and 2021.

### 2.2. Hoop House Studies

The GH, MV, and SK selections (Table 2) were the first set of tigernuts procured in 2008 from African markets in central NJ and evaluated in the hoop house at Rutgers Horticultural Farm 3 (RH3), 67 Ryders Lane, East Brunswick, NJ. These were followed in 2010 by selections NG1, NG2, NG3, and NG4 (Table 2) also procured in African markets in central NJ. Selections OG and T-USA (Table 2) purchased online from Organic Gemini, LLC (Brooklyn, NY, USA) and Tigernut USA, LLC (Hamilton, NJ, USA) were added to the germplasm in 2014 and 2019, respectively.

The 6 × 12 m^2^ hoop house used for growing tigernuts was covered with a clear double layer tube greenhouse film (6 mil gauge from Greenhouse Megastore), which was replaced every three years to sustain maximum sunlight penetration. Regulated heating units were provided to keep the hoop house above 18.3 °C during the cold months between November and May. The heating units were switched off between June and October when the ambient temperature ranged between 23.9 and 37.8 °C. Dehumidifiers were provided to regulate the relative humidity between 75 and 90% year-round. During the short daylight period between November and April, supplemental lighting was supplied using four regular incandescent lamps (Philips 65-Watt Incandescent BR30 Flood Light Bulb Soft White [2700 K]) arranged centrally approximately 2.4-m overhead and 3-m apart along the length of the hoop house. The lights were programmed to be on for 14-h per 24-h cycle. Plants were raised on 0.75-m tall 1.2 × 2.40-m^2^ metal benches arranged to allow free passage between benches for plant management and data collection. The hoop house was connected to the public water system for plant irrigation.

Tigernut plantlets (Figure 1) were raised in 48-hole black plastic flats filled with the Pro-Mix BX brand (Premier Pro-Mix BX 3.8CF BL White/Black #10380RG–Griffin (griffins.com) accessed on 27 January 2022). Both the plastic flats and the growth media were procured from the Griffin Greenhouse Supplies Company, Ewing, NJ, USA. A tuber was planted approximately 2.5 cm deep in each hole and watered adequately for four weeks before transplanting the plantlets into 20-cm diameter pots (20-cm tall), which were used for the comparative growth studies of tigernuts in the hoop house. The same media used for raising the tigernut plantlets was used for the plants grown in the 20-cm diameter pots. Peter’s soluble NPK 20-20-20 fertilizer (JR Peters Inc., Allentown, PA, USA) was applied as solution using 30 g/3.75 L (gallon) of water. Application was made at transplanting and at four and eight weeks after transplanting (WATP). At each time of fertilizer application, enough solution was applied until water started to drip at the bottom of the pot. Growth studies continued for 14–16 weeks to allow observation of plant growth habit and collection of foliage and tuber yield data. Data collected included the plant height, number of shoots per pot, foliage fresh and dry weight, number of tubers, tuber fresh weight, and the tuber yield ratio (TYR), which was defined as the tuber fresh weight divided by the foliage dry weight. Tuber distribution in the media was observed before harvesting manually using a 1/4th-inch wide wire screen to separate the tubers from the growth media. Fresh tuber weight was determined after cleaning the tuber under running water and open-air drying for 72–96 h under hoop house conditions. Foliage dry weight was determined by placing the freshly harvested foliage in a dryer set at 40 °C for 72–96 h.

### 2.3. Field Cultivation Studies

Tigernut may be planted in the field using the tuber directly or plantlets raised in the nursery. Due to significant variability previously observed in tuber sprouting and establishment in the field, the experiments reported in this paper were conducted using plantlets raised in hoop house nursery as described above. The use of plantlets allowed for the relatively uniform plant establishment and growth in the field. Annually, the tigernut nursery was initiated in mid- to late April to raise strong plantlets for transplanting in the field in late May to early June. Between 2014 and 2021, field evaluations of tigernut occurred in three locations in NJ, namely: (a) Rutgers Agricultural Research and Extension Center (RAREC), Bridgeton (southern NJ), (b) HF3, East Brunswick (central NJ), and (c) Specca Farms Pick-Your-Own, Bordentown (SPC-F) (central NJ). Previous observations at HF3 showed that tigernut performed much better under plastic mulch than on bare ground primarily because of suppression of weed interference, moisture conservation, and possibly temperature modulation. For this reason, the experiments reported in this paper were those conducted under black (BPM) and/or white-over-black (WPM) plastic mulch using approximately 4-week-old plantlets (or transplants) raised in the hoop house (Figure 1).

In 2014, eight tigernut selections, namely GH, MV, NG1, NG2, NG3, NG4, OG, and SK (Table 2) were compared at RAREC-and HF3-under BPM, and in 2020 and 2021 two selections, namely NG3 and T-USA were compared at SPC-F-under BPM and WPM mulches.

At all locations, soil tests were conducted prior to land preparation to determine the physical and chemical properties including texture, pH, organic matter (OM), N, P, K, Ca, Mg, and trace elements (Cu, Fe, Mn, Zn). Each year, between 15 and 30 May, soils were conventionally plowed and harrowed followed by fertilizer application (using the recommended rates based on soil test results), seedbed preparation (10–15 cm high and 75 cm wide at the top), and the laying of plastic mulch (black or white-over-black supplied by Griffin Farm Supplies, Ewing, NJ, USA) and trickle irrigation lines (also supplied by Griffin Farm Supplies, Ewing, NJ, USA). Distance from the center of one seedbed to the center of the next was 150 cm with a furrow of approximately 75 cm between the seedbeds. All operations were mechanized using standard farm machinery at the New Jersey Agricultural Experiment Station (NJAES).

Experimental plots were 1.5 m long and consisted of five tigernut plantlets spaced 30 cm apart along the center of the seedbed. A spacing of 90 cm separated one plot from the next along the seedbed row. Each year, between 30 May and 10 June, approximately 4-week-old tigernut plantlets were transplanted by placing them (and firmly covering them with soil) in holes made with a simple hand-pushed metallic auger, which drilled carefully through the plastic mulch. Moisture supply was primarily by natural rainfall supplemented with trickle irrigation only as needed.

Weeds that emerged around the tigernut stands were removed manually. In our studies, tigernut plants were relatively free of insect attack or disease interference. For this reason, no insect or disease control treatments were applied. Billbug (*Sphenophorus pavullus* Gyllenhal) infestations occurred sporadically in a few sites. This soil-borne grub attacked the young shoot at the growth point underground, too late in the season (≥8 weeks after transplanting) to allow any control treatments.

Plants were allowed to senesce and dehydrate before harvesting was initiated and this occurred between mid- to late October in NJ (about 14–16 weeks after planting). For data collection, three plants in the middle of the plot out of five in each experimental plot were harvested for foliar growth and yield determinations. After removing the plastic mulch, plants were dug up with a spade to collect 3–5 cm topsoil at the base of the plant and deposited onto a 1/4th-inch wide wire screen to manually separate tubers from the soil. This procedure was carried out when the soil was relatively dry and never done after a recent rainfall, which would make tuber separation from the soil very difficult (if not impossible). Plant foliage from the three harvested plants cut at ground level was dried in an oven set at 40 °C for 72–96 h for dry weight determination. Tubers were washed thoroughly in running water and air-dried in the hoop house for 72–96 h before measuring the tuber yield. Figure 2 shows the typical tuber drying process in our studies.

### 2.4. Experimental Designs and Statistical Analysis

For the hoop house studies, the completely randomized design (CRD) was used, and replications varied between four and six depending on propagule availability. The randomized complete block design (RCBD) was used for all the field studies with three or four replications depending on propagule availability. For both the hoop house and field experiments, data were analyzed using the IBM SPSS Statistics Grad Pack Premium V26.0 and treatment means were separated using the Tukey’s honestly significant difference (HSD) at 5% level of significance.

### 2.5. Tuber Analysis

Proximate and elemental analyses were carried out to compare the nutritional content of tigernut tubers under black and white plastic mulches at the SPC-F site, in 2020 and 2021. Air-dried tigernut tubers from each treatment (three replications each of NG3 and T-USA selections under black and white-over-black plastic mulch) were packaged carefully and shipped to Brookside Laboratories in New Bremen, Ohio (https://www.blinc.com/ accessed on 28 January 2022), for proximate and elemental analyses. The analytical protocols for digestible carbohydrates, digestible protein, dietary fiber, total fat, and macro- and micro-nutrients; and references are shown in Table 3. The data were analyzed using two subsamples from the composite of the three replications of each of the tigernut selections and used as two replications for each treatment.

## 3. Results

### 3.1. Hoop House Studies

Figure 3 shows the typical appearance of large- and medium-size tuber producing tigernuts (GH, MV, NG3, NG4, OG, SK, or T-USA) in the hoop house at 6 WATP when grown in 20-cm diameter pots using the BX brand of Pro-Mix. In other studies (not reported in this paper) NG1 and NG2 (which produce small tubers) developed narrower leaves and higher shoot densities than shown in Figure 3. Figure 4 shows the tigernut growth habit in the hoop house over eight weeks after planting (WAP). It was observed that tuber sprouts begin to emerge six to seven days after planting at 24–29 °C and continue over 14 days. Rhizome production by the primary shoot begins 3–4 weeks after planting the tigernut tuber when the primary shoot is at the fourth–fifth leaf stage. From 5–6 WAP rapid elongation of the rhizome occurred which may have resulted in the production of a secondary shoot or initiation of a tuber (Figure 4). Between seven and eight WAP, there was a rapid proliferation of secondary and tertiary shoots, and tubers at the milk stage. From 8–10 weeks tuber size and tuberization increased; more rhizomes and side shoots were produced giving rise to a higher number of shoots and tubers. By 12 WAP the oldest tubers reached maximum size and were mature, while the younger tubers continued to grow and develop. From 13–14 WAP, the foliage began to senesce and lodge. At this stage, most tubers attained maximum size and were ready for harvesting. At 16 WAP, foliage was dead, tubers were mature and ready for harvesting.

Figure 5 shows tigernut tuber distribution in the growth medium (Pro-Mix BX brand) in GH, MV, and SK selections at 12 WATP in the hoop house. Tubers were distributed throughout the 20-cm deep growth medium in the GH selection but limited to the upper 5 cm in MV and SK selections. This tuber distribution reflected rhizome growth habits in the three selections. In GH, extensive rhizome growth occurred before tuber initiation, whereas, in MV and SK, rhizomes were much shorter at the time of tuber initiation at the tip of the rhizome. Figure 6 shows tuber appearance in GH, MV, and SK tigernut selections. The GH and SK selections produced spherical (round) tubers while the MV selection produced oblong (elongated) tubers.

Table 4. shows the growth characteristics of GH, MV, and SK tigernut selections in the hoop house. At 12 WAP, GH at 91 cm tall was the shortest of the three selections. MV and SK were taller at >100 cm. Beyond 12 WAP, plant height no longer increased, but shoot number increased (data not shown). By 13 WAP, plants began to senesce and lodge. By 14 WAP, the foliage had turned 90–100% yellow and some leaves had turned brown. MV and SK produced a higher number of shoots and produced a higher shoot dry weight than GH (Table 4). However, tuber yield was equivalent in all three selections. The GH selection produced more tubers per shoot (six/shoot) and bigger tubers (2.8 g/tuber) than MV and SK selections. The GH selection also gave a higher tuber yield ratio (tuber yield/shoot dry wt.) than MV and SK. With these results, GH showed greater efficiency in tuber production than MV and SK selections in the hoop house.

### 3.2. Field Studies

Figure 7 shows the growth characteristics of tigernut selections in our collection under plastic mulch at 10 weeks after transplanting (WATP) at HF3. With a propagule spacing of 30 cm apart at the center of the seedbed, tigernut provided 100% seedbed cover at 10 WATP. Plastic mulch suppressed weeds satisfactorily, conserved moisture, and supported a vigorous plant growth on the seedbed. However, the furrows (inter-seedbed spaces) still needed adequate weed control to keep the plots clean and free of extraneous influences.

The combined data from experiments conducted between 2014 and 2016 at HF3 in East Brunswick, NJ, and RAREC, Bridgeton, NJ are shown in Figure 8 and Figure 9. In Figure 8 the NG3 selection produced the highest tuber yield (530 g from three stands) at the study locations in central and southern NJ, followed by SK, MV, NG4, NG2, and NG1 with 438, 410, 400, 386, 322 g from three stands, respectively. Selections OG and GH produced the lowest yields of <300 g from three stands (Figure 8).

Figure 9 shows the tuber yield ratio (TYR) from the tigernut selections compared in central and southern NJ between 2014 and 2016. The tuber yield ratio (TYR) defined as the tuber weight: shoot dry weight ratio, is a measure of tuber production efficiency. Selections NG3 and NG4 showed a superior TYR of 1.8 compared to the other selections (Figure 9). OG, MV, NG2, and SK were in the second tier with TYR between 1.0 and 1.2. GH and NG1 gave the lowest TYR (<0.8). The results of these experiments showed that NG3 and NG4 were more efficient in energy conversion into tuber production than the other selections under field conditions.

Table 5 shows the tuber yield from NG3 and T-USA selections under black (BPM) and white-over-black (WPM) plastic mulches at SPC-F in 2020 and 2021. Regardless of the type of mulch used, tigernut tuber yield was higher in 2021 than in 2020. The factor suspected to be responsible for the yield difference was the higher soil organic matter at the 2021 (2.8%) location than the 2020 location (1.6%), as other soil physical and chemical properties were similar. When summed over two years, tigernut performance under WPM was better than under BPM. The difference was more remarkable for the T-USA selection. The combined yield for 2020 and 2021 for NG3 selection under black plastic mulch was 3190 g from 10 tigernut stands compared to 3383 g under white-over-black WPM) (a 6% difference). For T-USA, the combined yield under BPM for 10 stands was 2295 g compared to 3505 g under WPM (a 52.7% difference). While the NG3 selection was less responsive to plastic mulch color, the T-USA selection showed a remarkable sensitivity, especially in 2021 (Table 5).

Among the food elements, the only observed effect of plastic mulch color occurred with total fat %, which was higher under WPM than BPM in the NG3 selection but similar in the T-USA selection (Table 6). Mulch type had no effect on the macronutrient content of tigernut tubers in both the NG3 and T-USA selections (Table 6). Among the micronutrients, the Fe level in the NG3 selection was higher under BPM in 2020 but slightly lower than under WPM in 2021. In 2021, Fe level was significantly higher under BPM in T-USA. In NG3 selection, Mn was higher in the tuber under WPM in 2020 and 2021. In the T-USA selection, Mn was much higher under BPM than under WPM. Mulch type had no effect on the Cu content of the tuber in both the NG3 and T-USA selections, but Zn was higher under BPM than under WPM in NG3 in 2020. Mulch color had no effect on Zn content in NG3 and T-USA tubers in 2021 (Table 6).

## 4. Discussion

### 4.1. Hoop House Studies

Growth studies on tigernut grown in Pro-Mix (BX brand) under hoop house conditions allowed a close look at the life cycle of tigernut to determine distinct growth stages. Four stages were identified, namely: (a) sprout emergence (6–14 days after planting [DAP]), (b) rhizome formation (3–4 WAP or fourth–fifth leaf stage), (c) tuberization (6–10 WAP), and (d) tuber maturation and foliage senescence (10–14 WAP). Stages b and c are similar to what has been reported for the weedy yellow nutsedge by Lingenfelter and Curran [24] and Schonbeck [25]. All the above- and below-ground growth had stopped at 16 WAP. Tuber distribution in the growth medium varied with tigernut selection. While tuber distribution was shallow in MV and SK, distribution was deep in GH. This distribution pattern was controlled by the extent of rhizome growth before the secondary shoot or tuber initiation. Rhizome growth was extensive in GH but limited in MV and SK. Two rhizome growth patterns were observed: (a) some rhizomes transformed into new shoots; and (b) others transformed into tubers. The principle (biological, chemical, or physical) that controls the rhizome terminal growth needs further investigation. Understanding this principle might help in the manipulation of tigernut rhizomes to produce more tubers or shoots as the situation demands, for a higher economic benefit. Foliage and tuber growth varied significantly among tigernut selections. In the hoop house GH selection with the least foliage growth produced as much tuber yield as MV and SK, which produced a significantly higher shoot number and weight (Table 4). This observation, which was expressed as tuber yield ratio (TYR = tuber fresh weight/shoot dry weight) showed that GH is more efficient at converting photosynthetic energy into tuber production than MV and SK in the hoop house. To the grower, this has implications, as most tigernut growers for tuber production would prefer a cultivar with higher TYR because of the reduced quantity of foliage to handle at harvesting. As reported by Asare et al. [34] and Maduka and Ire [2], tigernut tuber varies in size and color. Among the selections evaluated in the hoop house, NG1 was the only one with black color, the rest had brownish yellow color at harvest. MV was the only one with an oblong shape as shown in Figure 6. Other selections had a round (spherical) shape at harvest and became shriveled when dry, assuming a deeper brownish color and corrugated (crinkled) skin like the commercially available tuber.

### 4.2. Field Studies

The field experiments showed that all the tigernut selections evaluated in central and southern NJ completed their life cycle successfully when cultivated and managed between May and October. This confirmed that tigernut cultivation for the tuber is feasible in NJ and growers may take full advantage of this crop as an addition to the agricultural economy of the state. In Canada, Elford et al. [35] reported successful production of tigernuts under Ontario conditions. Added to the increasing popularity of tigernut production in Mexico [22,36], it seems north America (Canada, Mexico, and USA) now has the potential to contribute significantly to the global tigernut production in the future.

In this study, NG3 was the highest tuber yielding selection with 523 g from three tigernut stands. When converted to yield per hectare NG3 (at 523 g from three stands planted at 30 cm apart on rows that were 1.5 m apart) produced 3.7 mt/ha of tubers followed by SK (442 g) and MV (430 g) which produced 3.3 and 3.2 mt/ha, respectively. The other selections produced less than 3.0 mt/ha with GH (at 194 g) yielding the least at 1.4 mt/ha. The yields obtained in these studies compared favorably with the yields reported in Canada [34], Ghana [37], and Valencia [38]. The TYR placed NG3 and NG4 as the most efficient selections for the conversion of energy into tuber production under field conditions. Overall, NG3 gave the best performance in the field, combining high tuber yield (Figure 8) with high tuber production efficiency (Figure 9). For future commercial production considerations in central and southern NJ, NG3 ranked the highest among the eight selections compared in this study.

At SPC-F in Bordentown, NJ, tigernut production under WPM was superior to production under BPM. The T-USA selection demonstrated this more distinctly. The heat balance under the two mulches might have a role to play in these results. Excessive heat in the summer months especially during the active tuberization weeks between July and August might have reduced energy storage in the tuber under BPM, which has been reported to accumulate more heat than the WPM [39,40]. It is noteworthy that NG3 was less sensitive to mulch color. This biological variation in response as well as the higher tigernut tuber yield under WPM deserve further research, so future growers may take full advantage of the superior tuber yield associated with mulch color.

The results of tuber analysis for food elements, macro- and micro-nutrients indicated that tigernut tubers in NJ compare favorably with analytical results generally reported for tigernuts elsewhere [1,2,41]. Among the food elements, digestible carbohydrates ranged from 38–42%, digestible protein 6–8%, total fat 20–26%, and dietary fiber 7–8.6%. Potassium at 0.61–0.69% was highest in the tigernut tuber among the macronutrients, followed by P at 0.24–0.29% and Mg at 0.10–0.14%. Calcium was low (<0.05–0.07%). Among the micronutrients, Fe at 168–218 ppm (16.8–21.8 mg/100 g dry tuber) was the highest in tigernut tubers, implying that tigernut tuber is a good source of this element in human nutrition. Tigernut has been reported to have an affinity for nutrient accumulation in the tuber [42,43] and Fe has been found to be one of the metals. It has been measured in raw tuber and the flour and different groups have reported different values, ranging from 6.5–41 ppm in the flour [44,45,46]. In this study the level of Fe detected in the tuber (168–218 ppm) was much higher than previously reported. Fe was followed by Zn at 39–50 ppm. Manganese content varied widely from 6–68 ppm, while Cu was low at 6–11 ppm. Mulch color did not show any remarkable impact on tigernut tuber nutritional content. Among the food, elements total fat % was the only element that seemed higher under WPM in NG3 selection, but mulch color had no impact in T-USA. Mulch color had no impact on macronutrient content in tigernut tuber. Iron level in tigernut tuber was high in the plastic mulch treatments in 2020 and 2021 but showed no consistent trend. Zinc level (50 ppm) was higher under BPM than WPM in 2020 in NG3 but remained similar in both NG3 and T-USA under the two mulch colors in 2021. Manganese was higher under WPM than under BPM in NG3 but the reverse was observed with T-USA. Mulch color did not affect Cu level in tigernut tubers. These observations suggest that mulch color has a limited (but important) effect on nutrient content in tigernut tubers. Literature is scarce to corroborate what was observed in this study and more work is needed to define clearly how mulch color might impact tigernut tuber quality in NJ.

## 5. Conclusions

Studies conducted between 2008 and 2021 on tigernut selections in NJ provided an understanding of the growth habit of this plant in the hoop house and adaptation to and productivity in NJ agroecosystems. It was confirmed that tigernut is adapted to NJ and may be commercialized with resultant tuber quality comparing favorably with the tubers imported into the United States from West Africa, Spain, and Mexico. It was also confirmed that yield was impressive under good management practices, especially under black or white-over-black plastic mulch. The high Fe and Zn contents in the tuber are important selling points for the tigernut tuber in the United States to supplement diet intake as the population manages the COVID-19 pandemic and other nutrition-related diseases. In other unpublished work, the tigernut tuber in all the selections studied did not survive the cold winter months of December to March in the field and therefore the plant poses no threat of becoming an invasive weed in Northeastern United States agroecosystems. Growers in NJ have an opportunity to become active players in the expanding tigernut global market, currently estimated at over two billion dollars; and in the United States with import value estimated at >$660 M in 2020 [22].

## 6. Suggested Research Areas

(a) To fast-track tigernut adoption in NJ and the United States, developing an efficient harvesting technology at the small/medium farmer level is imperative. Currently, manual harvesting is the most labor-intensive component of production practices, and this poses the greatest obstacle to adoption by growers. Labor cost for harvesting reduces the profitability potential significantly. This operation must be mechanized for the NJ grower to be willing and able to engage in profitable commercial tigernut production. (b) The impact of mulch color on tuber quality needs further investigation to enhance recommendations to growers. (c) Physiologically, the principle that controls tuber production in tigernuts also needs to be better understood for maneuverability to achieve higher tuber yield and greater TYR. (d) Billbug is a potential insect pest that must be controlled to ensure optimum tuber yield and quality.

## Figures and Tables

**Figure 1 plants-11-00897-f001:**
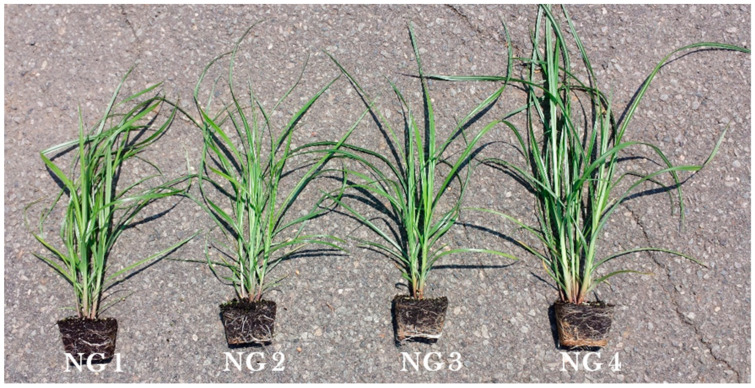
Tigernut plantlets at four weeks (30 days) after planting the tuber in Pro-Mix BX brand media in 48-hole flats under hoop house conditions and fertilized with NPK 20-20-20 solution (30 g/gallon or 3.75 L of water) at 15 days after planting. The NG selections originated from Nigeria and varied in size from small (NG1) to large (NG4). (Photo: Albert Ayeni).

**Figure 2 plants-11-00897-f002:**
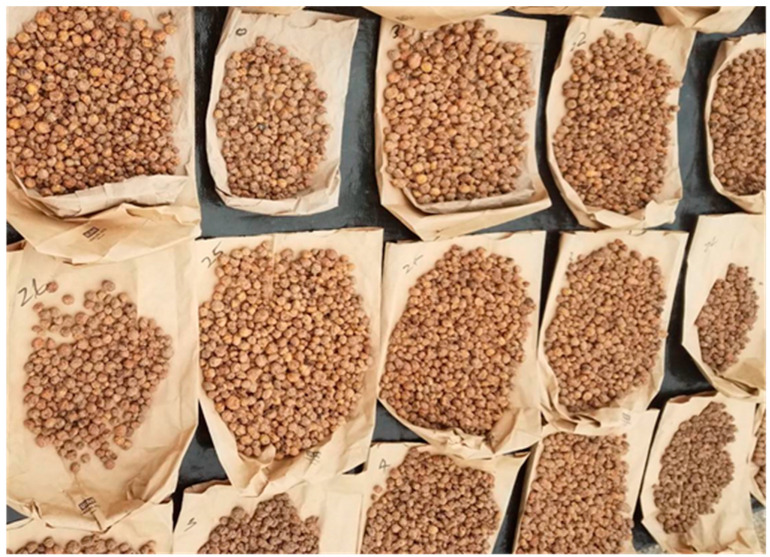
Air drying of freshly harvested tigernut tubers in the hoop house. Low humidity and free air circulation facilitate the drying process and preserve tuber quality. Photo: Albert Ayeni.

**Figure 3 plants-11-00897-f003:**
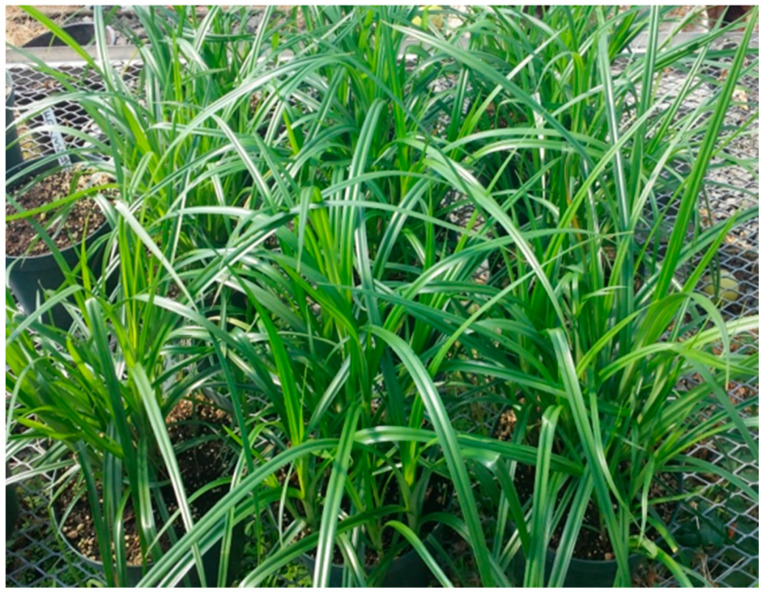
Typical appearance of tigernut plant in 20-cm diameter pots grown in Pro-Mix (BX brand) in the hoop house at 6 weeks after transplanting (Photo: Albert Ayeni).

**Figure 4 plants-11-00897-f004:**
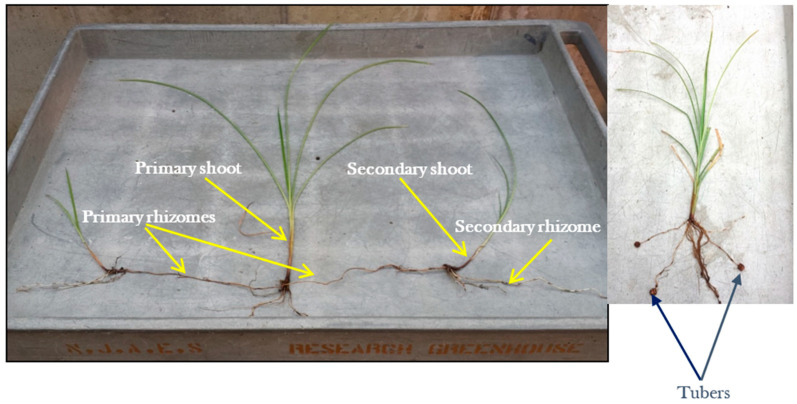
Tigernut growth habit (Photos: Albert Ayeni).

**Figure 5 plants-11-00897-f005:**
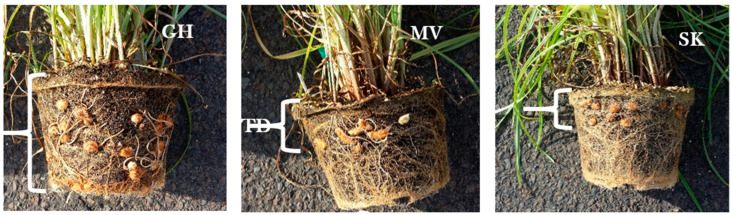
Tuber distribution in growth medium (Pro-Mix BX brand) at 12 weeks after transplanting in the GH, MV, and SK tigernut selections in the hoop house (See Table 1 for a full description of acronyms). The enclosure symbol shows the depth of tuber distribution (TD) in the growth medium (Photo: Albert Ayeni).

**Figure 6 plants-11-00897-f006:**
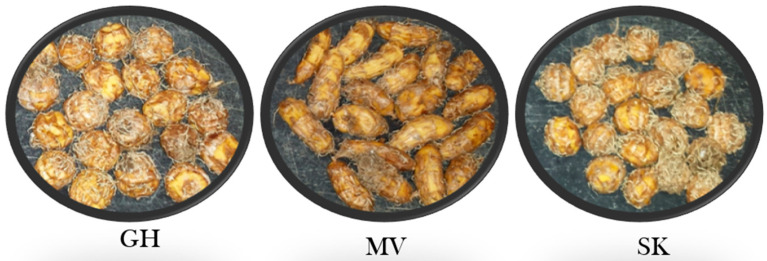
Tuber size and shape in the GH, MV, and SK tigernut selections grown in the hoop house using Pro-Mix BX brand. Tubers were harvested 14 weeks after planting (See Table 1 for a full description of acronyms) (Photos: Albert Ayeni).

**Figure 7 plants-11-00897-f007:**
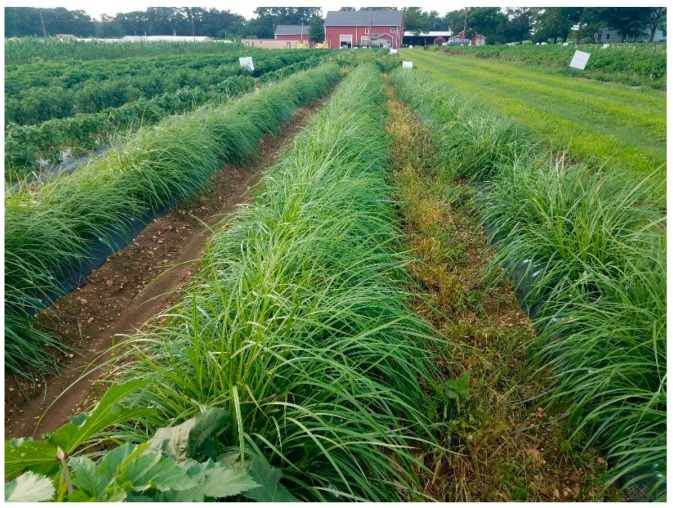
Tigernut growth under black plastic mulch at 10 weeks after transplanting at Horticulture Farm 3, East Brunswick, NJ (Photo: Albert Ayeni).

**Figure 8 plants-11-00897-f008:**
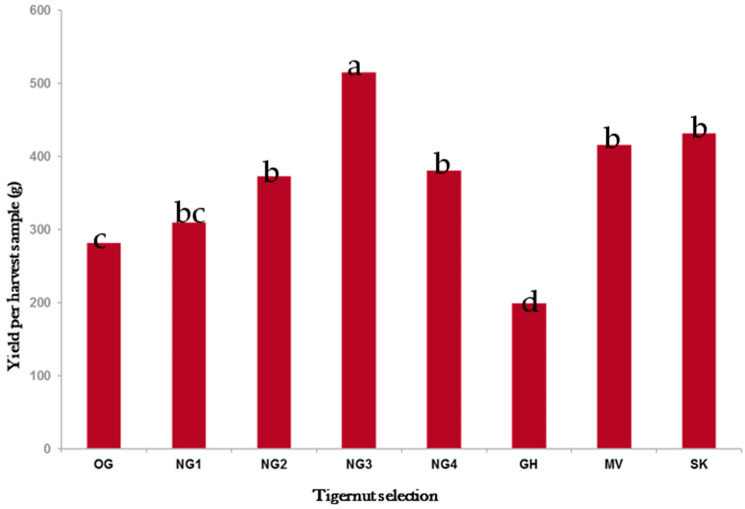
Combined tuber yield (g) in tigernut selections under black plastic mulch at Horticulture Farm 3, East Brunswick, Central NJ, and Rutgers Ag Research and Extension Center, Bridgeton, Southern NJ Bars show treatment means separated using the Tukey’s HSD at 5% significance level. Bars with similar letters are not statistically different.

**Figure 9 plants-11-00897-f009:**
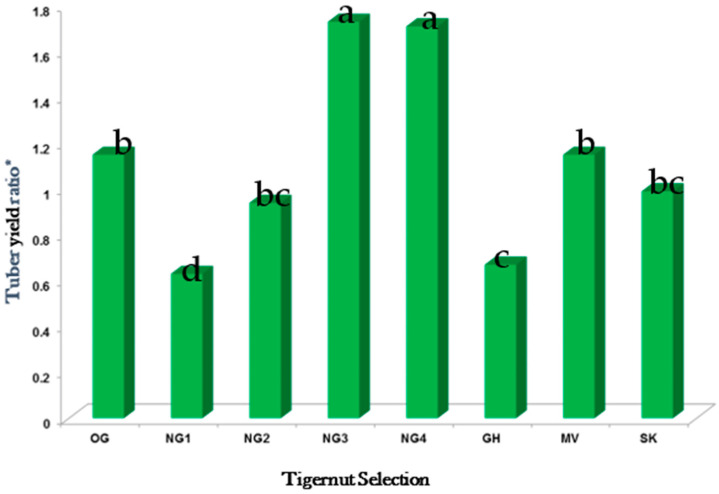
Combined results for tuber yield ratio in tigernut selections grown at Horticulture Farm 3, East Brunswick, Central NJ, and Rutgers Ag Research and Extension Center, Bridgeton, Southern NJ * Tuber yield ratio is the total fresh tuber weight divided by the foliage dry weight. Bars show treatment means separated by Tukey’s HSD test at a 5% significance level. Bars with similar letters are not statistically different.

**Table 1 plants-11-00897-t001:** Tigernut importation share and importation value in 2020; and five-year growth in import value among the top 10 importing countries (Source: Tridge—https://www.tridge.com/intelligences/tiger-nut/import, accessed on 27 January 2022).

Country	Import Share (%) (2020)	Import Value (US$M) (2020)	Five-Year Growth in Import Value (%) (2015–2020)
USA	30.68	660.48	26.52
China	16.51	355.47	4.02
Germany	8.67	186.73	28.77
Mexico	6.9	148.53	68.57
Italy	6.47	139.23	52.52
Netherlands	3.84	82.77	16.26
Canada	3.62	78.03	2.18
UK	3.15	67.74	11.44
France	2.42	52.1	31.86
Spain	2.26	48.72	36.51
		Average Five-Year Growth in Import Value	27.90%

**Table 2 plants-11-00897-t002:** Tigernut selections used for hoop house and field studies at Rutgers University between 2008 and 2021.

Tigernut ID	Tuber Description	Source	Year First Evaluated
NG1	Black and small size round tubers (<5 mm diameter)	African Market in Central NJ, imported from Nigeria	2010
NG2	Brownish-yellow and small round tubers (<5 mm diameter)	African Market in Central NJ, imported from Nigeria	2010
NG3	Brownish-yellow large/medium size round tubers (5–15 mm diameter)	African Market in Central NJ, imported from Nigeria	2010
NG4	Brownish-yellow large round tubers (7.5–15 mm diameter)	African Market in Central NJ, imported from Nigeria	2010
GH	Brownish-yellow large round tubers (10–15 mm diameter)	African Market in Central NJ, imported from Ghana	2008
MV	Brownish-yellow large/medium oblong tubers (up to 20 mm long, 5–10 mm wide at the center)	African Market in Central NJ, imported from Monrovia, Liberia	2008
OG	Brownish-yellow large/medium round tubers (5–15 mm diameter)	12 oz Commercial package from Organic Gemini, Brooklyn, NY, USA	2014
SK	Brownish-yellow large/medium round tubers (5–15 mm diameter)	African Market in Central NJ, imported from Suakoko, Liberia	2008
T-USA	Brownish-yellow large/medium round tubers (5–15 mm diameter)	12 oz Commercial package from Tigernut USA, Hamilton, NJ, USA	2019

**Table 3 plants-11-00897-t003:** Tigernut tuber analytical methods for digestible carbohydrates, digestible protein, dietary fiber, total fat, and elemental macro-and micro-nutrients (based on dry matter samples).

Plant Nutrient	Analytical Protocol	Reference
Digestible Protein	AOAC 990.03 Combustion analysis by Elementar Max Exceed	[30]
Dietary Fiber	AOAC 962.09 crude fiber by filter method	[31]
Total Fat	AOAC 920.39 crude fat by ether extract	[32]
Macro and Micro Elements	Modified AOAC 985.01, metals by ICP determination (digestion by nitric acidin Teflon vessels digested in a CEM Mars Express microwave)	[33]
Digestible Carbohydrates	Calculated	[32]

**Table 4 plants-11-00897-t004:** Growth characteristics of GH, MV, and SK selections of tigernut in 20-cm diameter pots in the hoop house. The study was terminated 14 WAP when vegetative growth had stopped.

Tigernut Selection *	Plant Height (cm) (12 WAP ^1^)	Number of Shoots (14 WAP)	Shoot Dry wt. (g) (14 WAP)	Number of Tubers (14 WAP)	Tuber Yield (g)	Tuber #/Shoot	Tuber Size (g)	Tuber Yield Ratio ^2^ (TYR)
GH	91 b **	13 b	31 b	28 b	78.6 ns	6.0 a	2.8 a	2.5 a
MV	104 a	21 a	39 a	100 a	76.0 ns	3.6 b	0.8 b	1.9 b
SK	111 a	20 a	41 a	93 a	68.0 ns	3.4 b	0.7 b	1.7 b

^1^ WAP = Weeks after planting; ^2^ Tuber yield ratio (TYR) = Tuber yield/Shoot dry wt. * See Table 1 for a full description of acronyms. ** Within a column, figures followed by similar letters are not significantly different (Tukey’s HSD test at 5% level), ns = no significant difference.

**Table 5 plants-11-00897-t005:** Effect of black and white-over-black plastic mulches on tuber yield in the NG3 and T-USA tigernut selections at SPC-F in 2020 and 2021.

Tigernut ID	Black Plastic Mulch (BPM)		White-over-Black Plastic Mulch (WPM)	
	2020	2021	2020	2021
	Tuber Fresh wt. (g) from Five Stands			
NG3 *	1550 a **	1640 a	1243 a	2140 b
T-USA	855 b	1440 b	825 b	2680 a

* Please see Table 1 for a full description of acronyms ** Within column, different letters after the figures show significant difference using Tukey’s HSD test at 5% level.

**Table 6 plants-11-00897-t006:** Proximate and elemental analyses for NG3 and T-USA tigernut selections harvested at SPC-F in 2020 and 2021.

Analysis	NG3 *				T-USA	
	Black Plastic Mulch	WhiteOB ** Plastic Mulch	Black Plastic Mulch	WhiteOB Plastic Mulch	Black Plastic Mulch	WhiteOB Plastic Mulch
	2020		2021			
Digestible carbohydrate (%)	41.3 ns	38.3 ns	41.9 ns	37.6 ns	38.9 ns	40.0 ns
Digestible protein (%)	8.1 ns	7.3 ns	7.5 ns	7.2 ns	7.1 ns	6.4 ns
Fat (%)	21.6 b	25.2 a	20.0 b	26.4 a	25.6 a	24.4 a
Dietary fiber (%)	7.2 ns	8.3 ns	8.6 ns	8.1 ns	8.0 ns	7.6 ns

Potassium (K) (%)	0.67 ns	0.65 ns	0.69 ns	0.61 ns	0.64 ns	0.66 ns
Calcium (Ca) (%)	<0.05 ns	0.07 ns	<0.05 ns	<0.05 ns	0.06 ns	0.05 ns
Phosphorus (P) (%)	0.27 ns	0.29 ns	0.24 ns	0.27 ns	0.25 ns	0.24 ns
Magnesium (Mg) (%)	0.11 ns	0.14 ns	0.10 ns	0.12 ns	0.13 ns	0.12 ns

Iron (Fe) (ppm)	198 a	138 b	208 b	218 b	243 a	168 c
Manganese (Mn) (ppm)	8 b	32 a	6 c	24 b	68 a	29 b
Copper (Cu) (ppm)	9 ns	11 ns	8 ns	9 ns	9 ns	6 ns
Zinc (Zn) (ppm)	50 a	42 b	44 ns	41 ns	43 ns	39 ns

Statistical analysis compared plastic mulch means for each analytical item within the year, and was separated using Tukey’s HSD test at 5% significance level; letter “a” or “b” or “c” shows significant difference between or among the compared means; ns = no significant difference. * See Table 1 for a full description of acronyms. ** WhiteOB = White-over-black.

## Data Availability

All data provided in this article were internal to the research project.

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
