# Peer review of "Hoop House and Field Evaluation of Tigernut (Cyperus esculentus L. var. sativus Boeck) Selections in New Jersey, USA"

_plants, 2022, doi:10.3390/plants11070897_

Round 1

Reviewer 1 Report

The work deals with quite interesting issues regarding the hoop house and field evaluation of tigernut selections in New Jersey, USA. The article presents original and long-term research material. However, the revision is still needed before the acceptance of this manuscript.
The titles of Figures are described in very widely form. The inscriptions below the pictures should be short and clear. I would suggest some information move to the text.
Figure 2: The title is not text. Please correct it.
My suggestion, if the figure consists of a few pictures, each of them should be explained.

Author Response

Reviewer #1

Comments and Suggestions for Authors

The work deals with quite interesting issues regarding the hoop house and field evaluation of tigernut selections in New Jersey, USA. The article presents original and long-term research material. However, the revision is still needed before the acceptance of this manuscript.
The titles of Figures are described in very widely form. The inscriptions below the pictures should be short and clear. I would suggest some information move to the text. Comment has been addressed as necessary. Author prefers to leave some titles unedited where they help the reader to understand the figure better.
Figure 2: The title is not text. Please correct it. Title font and size for Figure 2 have been modified to conform with the text
My suggestion, if the figure consists of a few pictures, each of them should be explained. All figures have been fully described to reflect the photo contents.

Reviewer 2 Report

I was pleased to read how a smaller crop like Tigernut can be valued in a new environment and not just considered a weed. The experimental work needs a thorough revision in all its parts
- Line numbering missing in much of the text.
- Line 4. Lack of asterisk or number to indicate affiliation and correspondence.
- Line 19. It results in a cut word “top”.
- Line 142. The figure could be inserted in the results in order to argue for the development of plants after sowing.
- Line 205. The characters should be the same as the text.
- Line 206. Lack of indication of the software used for statistical analysis.
- Line 228. The text speaks of “typical morphology… ..” but the figure refers to plants grown in pots. Please correct.
- It would be appropriate to report the acronym in full for a simpler and clearer reading (Fig. 5.6, Tab. 4).
- Line 284. In the caption of the table it would be appropriate to insert the words "means followed by the same letter in the same column are not significantly different according the Tukey's test at 5% level of significance" or something similar would be fine.
- Figure 8 should be explained in the text and as for Figure 9 it would be appropriate to enter the standard deviation in the columns.
- The comment of figure 6 seems a bit too poor given the large amount of data shown, it would be appropriate to deepen in the text.
- In the discussion due to the lack of line numbering it is difficult to make suggestions.
- In the last sentence of paragraph 4.1 indicate which studs, it is advisable to insert the bibliography or delete the reference.
- Section 4.2 should be revised because the data should be reported in the results section.
- The discussion paragraph should discuss the findings and how they can be interpreted in perspective of the previous studies and working hypotheses.
- Discussions lack bibliographic references, need to be completely revised.

Author Response

Reviewer #2

Comments and Suggestions for Authors

I was pleased to read how a smaller crop like Tigernut can be valued in a new environment and not just considered a weed. The experimental work needs a thorough revision in all its parts
- Line numbering missing in much of the text. Editor’s help is requested to correct this problem
- Line 4. Lack of asterisk or number to indicate affiliation and correspondence. Done
- Line 19. It results in a cut word “top”. Agreed

- Line 142. The figure could be inserted in the results in order to argue for the development of plants after sowing. Author prefers to keep Figure 1 where it is as part of the Materials and Methods for the studies. It is not a result of the reported studies, but an essential material used for the studies.

- Line 205. The characters should be the same as the text. Done
- Line 206. Lack of indication of the software used for statistical analysis. We used IBM SPSS Statistics Grad Pack Premium V26.0, this has been inserted paragraph 2.4.
- Line 228. The text speaks of “typical morphology… ..” but the figure refers to plants grown in pots. Please correct. Morphology has been changed to appearance and in Figure 3 title has been modified.
- It would be appropriate to report the acronym in full for a simpler and clearer reading (Fig. 5.6, Tab. 4). “See Table 1 for a full description of acronyms” has been inserted. Reporting acronyms in full will be too wordy for the Figures and Table
- Line 284. In the caption of the table it would be appropriate to insert the words "means followed by the same letter in the same column are not significantly different according the Tukey's test at 5% level of significance" or something similar would be fine. Done
- Figure 8 should be explained in the text and as for Figure 9 it would be appropriate to enter the standard deviation in the columns. Figure 8 is explained in Line 322-326; rather than use standard deviation of means, author prefers to retain the use of Tukey’s HSD test to separate the means in both Figures 8 and 9.
- The comment of figure 6 seems a bit too poor given the large amount of data shown, it would be appropriate to deepen in the text. This comment is not clear, Figure 6 is a photo showing tuber shapes and sizes for different tigernut selections. Author needs some clarification to respond better to this reviewer’s comment.
- In the discussion due to the lack of line numbering it is difficult to make suggestions. Author needs the Editor’s help to insert lines to the text from pages 12-15
- In the last sentence of paragraph 4.1 indicate which studies, it is advisable to insert the bibliography or delete the reference. Sentence is deleted.
- Section 4.2 should be revised because the data should be reported in the results section. The data collected from the field studies were reported in the results section. In the discussion, yield data were converted to yield per hectare for the purpose of comparing the yield from this study with tigernut tuber yield results reported elsewhere around the world.
- The discussion paragraph should discuss the findings and how they can be interpreted in perspective of the previous studies and working hypotheses. This is precisely what was done. It should be noted that relevant bibliographic references were cited as necessary and in the relevant areas. The focus of this study was to highlight the core findings that would benefit growers and consumers of tigernut tubers in NJ and the United States. For this reason, discussions regarding the biological and morphophysiological perspectives of tigernut plant and tuber were minimized. 
- Discussions lack bibliographic references, need to be completely revised. Discussions have been revised substantially with bibliographic references.

Round 2

Reviewer 2 Report

Dear authors, I am glad that you have considered many of my suggestions, some of your answers could have been undertaken differently but the explanations provided were comprehensive.